# Mechanisms by Which SARS-CoV-2 Invades and Damages the Central Nervous System: Apart from the Immune Response and Inflammatory Storm, What Else Do We Know?

**DOI:** 10.3390/v16050663

**Published:** 2024-04-24

**Authors:** Zihan Sun, Chunying Shi, Lixin Jin

**Affiliations:** 1Qingdao Medical College, Qingdao University, Qingdao 266071, China; 2Department of Human Anatomy, Histology and Embryology, School of Basic Medicine, Qingdao University, Qingdao 266071, China

**Keywords:** COVID-19, SARS-CoV-2, viral infection, CNS, neurology

## Abstract

Initially reported as pneumonia of unknown origin, COVID-19 is increasingly being recognized for its impact on the nervous system, despite nervous system invasions being extremely rare. As a result, numerous studies have been conducted to elucidate the mechanisms of nervous system damage and propose appropriate coping strategies. This review summarizes the mechanisms by which SARS-CoV-2 invades and damages the central nervous system, with a specific focus on aspects apart from the immune response and inflammatory storm. The latest research findings on these mechanisms are presented, providing new insights for further in-depth research.

## 1. Introduction

Corona Virus Disease 2019 (COVID-19), caused by SARS-CoV-2, has made a significant impact on global public health, economic development, and social stability over the past three years. COVID-19 has been associated with various neurological symptoms, including cognitive impairment, sleep disturbance, anosmia, and possible mental illness [1,2]. Additionally, there have been numerous reports and epidemiological studies on the symptoms and sequela of the nervous system [3,4]. As the pandemic continues, research efforts are increasingly being focused on understanding the impact of SARS-CoV-2 on the nervous system.

SARS-CoV-2 is not the first coronavirus suspected of infecting the nervous system. SARS was previously reported to infect neurons via human angiotensin-converting enzyme 2 (ACE2) [5,6], and proinflammatory cytokines and chemokines were upregulated in the lungs and brain following infection [7]. There are case reports revealing that Middle East Respiratory Syndrome (MERS) patients had a severe nervous system syndrome [8]. Dipeptidyl peptidase 4 (DPP4) was determined to be a functional receptor of MERS [9], and experiments demonstrated that the death of MERS-infected mice was related to brain infection. MERS has also been confirmed to replicate in human central nervous system (CNS) cell lines, primary porcine astrocytes, and mouse astrocytoma cell lines [10]. In addition, HCoV-OC43 and HCoV-229E have been reported to have the potential to carry out nerve invasion [11]. HCoV-OC43 can induce postinfectious necrotic apoptosis (RCD) through Receptor-Interacting Protein Kinase 1 (RIP1) and Mixed Lineage Kinase Domain-Like (MLKL) [12], leading to neuropathology and motor deficits [13]. Another study suggested that the passage of HCoV-OC43 from the nasal cavity to the olfactory bulb, piriform cortex, and brainstem might be caused by axonal propagation [14]. HCoV-229E has been poorly studied, and in vitro studies have shown that astrocytes, oligodendrocytes, and neuronal cell lines are all susceptible to infection [15]. In general, SARS-CoV-2 is not the only coronavirus that can infect the CNS; pathological studies and experimental evidence of several coronaviruses have demonstrated that central nervous system infection can occur. However, it must be emphasized that whether via SARS-CoV-2 or other reported neurotropic viruses, CNS infection is still a rare event overall [16], despite a number of reports of CNS symptoms.

The genome of SARS-CoV-2 is approximately 29.9 kb in length. The first part of its genomic RNA encodes non-structural proteins (NSPs), which can be directly translated into two polyproteins, pp1a and pp1b. The second part mainly encodes four structural proteins: spike (S), membrane (M), envelope (E), and nucleocapsid (N). Extensive research has been conducted on the cellular infection process used by SARS-CoV-2. This virus’s S protein recognizes host cell surface receptors and binds to the cell membrane, allowing it to penetrate and release viral genomic RNA into the cell’s interior. The virus hijacks host ribosomes to transcribe two viral replicase polyproteins. These polyproteins are then processed into 16 mature NSPs by the virus’s main protease (M^pro^) and papain-like protease (PL^pro^). The NSPs can assemble into replication and transcription complexes (RTC), initiating viral RNA replication and transcription. Finally, RNA assembles with structural proteins into mature daughter virions, which are released through exocytosis [17].

ACE2 has been identified as the primary receptor of SARS-CoV-2 [18]. Other receptors or factors associated with the CNS include C-type lectins, DC-SIGN, L-SIGN, CD147, and Neuropilin-1 (NRP1). Additional relevant analyses involve transmembrane protease serine subtype 2 (TMPRSS2) and cathepsin L [19]. Human astrocytes and pericytes express certain receptors [20], which might enable a direct attack on nerve and glial cells via SARS-CoV-2 rather than solely affecting the CNS through immune response and inflammation [21,22]. Pathological studies on deceased COVID-19 patients have also revealed the presence of this virus in non-respiratory tissues, including the brain, even in the absence of obvious symptoms [23]. An experiment was conducted in which SARS-CoV-2 was used to infect K18-hACE2 mice that had undergone transfer of the human ACE2 gene and k18 promoter. This study found evidence of nerve invasion by SARS-CoV-2 in a portion of the deceased mice’s brains [24]. Therefore, it is crucial to understand how COVID-19 causes CNS symptoms while considering SARS-CoV-2′s nature as a neuro-invasive virus. Although some mechanisms have been proposed, our understanding of SARS-CoV-2’s effects on the nervous system remains incomplete. This review summarizes recent research on viral mechanisms that invade and damage the CNS with a specific emphasis on aspects beyond immune responses and inflammatory storms and suggests promising future research directions within a reasonable scope.

## 2. The Pathway of SARS-CoV-2’s Invasion of the CNS

There are two potential routes through which SARS-CoV-2 can enter the CNS: crossing a barrier, including the blood–brain barrier (BBB) or the blood–cerebrospinal fluid barrier (B-CSF-B), or nerve invasion. Figure 1 illustrates possible invasion pathways.

### 2.1. Crossing BBB or B-CSF-B

SARS-CoV-2 was found to cross the BBB through a transcellular pathway instead of via disrupting tight junctions (Figure 1A). In vitro cell experiments confirmed viral infection and replication in brain microvascular endothelial cells, although there was no cytopathic effect. The experiments also demonstrated BBB crossing without compromising tight junction integrity [25]. In vivo animal experiments have demonstrated that the S1 protein crosses the BBB and peripheral tissue through adsorptive transcytosis mechanisms after an intravenous injection of radioactive-iodine-labeled S1 protein. The process of crossing the BBB was found to be mediated by ACE2 [26]. A recent study examined the susceptibility of induced pluripotent-stem-cell-derived brain capillary endothelial-like cells (BCECs) to SARS-CoV-2 infection. This study suggests that there is a theoretical possibility that the virus can enter the CNS through the BBB [27]. The virus was found to be taken up at the luminal side of a transwell model, a process followed by intracellular replication, cross-cellular transport, and the release of viral particles at the basolateral side. Although these simulations demonstrated how this virus traverses the blood–brain barrier (BBB), direct evidence is lacking regarding this process in humans. Furthermore, anti-S protein, anti-ACE2, anti-NRP1, or the TMPRSS2 inhibitor nafamostat reduced SARS-CoV-2 entry into the brain capillary endothelium-like cells [27], suggesting the potential involvement of invasion recognition molecules.

The situation for the B-CSF-B is similar to that of the BBB. However, there is increasing evidence suggesting an impact on the choroid plexus epithelium (Figure 1B).

Laura Pellegrini et al. explained the mechanism by which SARS-CoV-2 infects the choroid plexus, focusing on receptor expression and demonstrating its ability to disrupt the B-CSF-B in human organoids [28]. These researchers confirmed that ACE2 and TMPRSS2, which are expressed exclusively in mature lipoprotein-expressing choroid plexus cells, serve as entry factors for SARS-CoV-2 infection. In contrast, neurons or neuronal progenitors did not express these molecules targeted by the virus invasion. Pseudo-virus infection of brain organoids subsequently revealed selective targeting of choroid cells [28]. The expression of factors such as ACE2, which is only present on the epithelial cells of the choroid plexus of the organoids, may control susceptibility to SARS-CoV-2 infection. The cited experiment’s results suggest that this virus may enter the CNS by infecting epithelial cells. Additionally, the researchers observed that the blood-borne SARS-CoV-2 infiltrated cells on the basal side, and these cells were characterized by a lower expression of ACE2. This observation is consistent with this virus’s relative localization within the bloodstream. Therefore, both in vitro experimental phenomena and in vivo physiological structures support the idea that this virus infects epithelial cells to enter the CNS. However, further animal experiments are still necessary in order to confirm this process in vivo.

Another study comparing the susceptibility of the BBB and B-CFS-B to COVID-19 found that SARS-CoV-2 infected the epithelial cells of the B-CFS-B but not the endothelial cells or pericellular cells of the BBB [29]. The authors of this study utilized human-induced pluripotent stem cells to demonstrate higher expression of ACE2 and TMPRSS2 in choroid plexus epithelial cells. In contrast, the endothelial cells and pericytes of the BBB lacked sufficient relevant receptors, indicating relatively greater vulnerability to B-CFS-B infection. In addition, this study found that SARS-CoV-2 infection was more likely to occur in outer basement-membrane-facing cells of human choroid plexus papilloma (HIBCPP) when exposed to this virus, indicating that viral infiltration originated from the bloodstream [29]. The presence of S protein transcripts in the patient’s choroid plexus epithelial cells supported this conclusion. The study suggests that choroid epithelial cells are more susceptible to SARS-CoV-2 infection than BBB endothelial cells or pericytes.

In summary, the current findings suggest that SARS-CoV-2 might primarily invade the CNS by crossing the B-CSF-B, although there is also copious evidence of its ability to breach the BBB. It is important to note that inflammation caused by SARS-CoV-2 may contribute to BBB disruption [30]. Further investigation is needed to determine whether this barrier breakdown signifies the initiation of SARS-CoV-2’s entry into the CNS or if it is just one aspect of CNS disturbance.

### 2.2. Nerve Invasion

SARS-CoV-2 may invade the brain through nerve invasion, in addition to crossing the BBB or the B-CSF-B. Extensive research has focused on the olfactory nerve mucosa as a potential route for SARS-CoV-2 invasion into the CNS. Viral particles and RNA have been detected in both the olfactory mucosa and neuroanatomical regions associated with olfactory projection [31]. Olfactory cells are located in the mucosa of the superior nasal concha and the upper part of the nasal septum, and their central process passes through the hard meninges to reach the olfactory bulb. Then, fibers from olfactory bulb neurons terminate in the olfactory center (the temporal uncinate gyrus, anterior hippocampal gyrus, and amygdala). This anatomical structure of olfactory neurons suggests that SARS-CoV-2 could infect neurons in the olfactory mucosa and use axonal transport for neural invasion (Figure 1B). In another study, K18-hACE2 mice infected with SARS-CoV-2 were used to observe brain damage chronologically. According to this study, SARS-CoV-2 immune markers were initially detected only in the olfactory mucosa 2 days after inoculation. However, by day 4, these markers had spread to most brain regions. The maximum distribution of the virus throughout the brain was observed at days 6 to 7, indicating that the infection starts in the olfactory mucosa and progresses into the CNS [32]. Furthermore, an autopsy report indicated that nerve damage caused by SARS-CoV-2 decreases in severity whilst moving from the olfactory nerve to the gyrus rectus (located at the most medial margin of the inferior surface of the frontal lobe) and brainstem, which supported the hypothesis of nerve invasion through the olfactory mucosa [33]. Additionally, previous studies have examined other cranial nerves and identified the entry site of SARS-CoV-2 in the structural elements of human glossopharyngeal and vagus nerves [34]. In vitro experiments have also confirmed that SARS-CoV-2 can transmit anterogradely or retrogradely along the axon [35]. As the primary neural structure exposed to SARS-CoV-2 infection, the olfactory nerve is considered a potential port of CNS invasion for this virus, and other nerves may also serve as invasion pathways. Evidence from studies on the presence of viruses in the CNS and the process of nerve invasion supports this view. Current studies suggest that SARS-CoV-2 may enter the CNS via neuroinvasion, although the morphological detection of individual viral particles in axons is challenging.

However, there is evidence that contradicts the notion of an invasion path through the olfactory nerve. Autopsies of COVID-19 patients revealed that fibroblasts surrounding the olfactory nerve formed an anatomical barrier against SARS-CoV-2 invasion. In all autopsy cases, it appeared that one or more layers of fibroblasts surrounded the olfactory nerve to prevent infection. Additionally, they found that the virus did not invade the olfactory bulb or the matter of the frontal lobe [36], suggesting that SARS-CoV-2 may not infect olfactory nerve cells. While neuroretrograde transmission is theoretically possible, more experimental evidence is needed to confirm this route of SARS-CoV-2 invasion.

Other potential mechanisms for neurotropic virus invasion include transmission through peripheral nerves (e.g., poliovirus, adenovirus, herpes virus, and rabies virus) or by using infected white blood cells as a Trojan horse to cross the BBB (HIV and Zika virus) [37]. These pathways may also be used by SARS-CoV-2 to invade the CNS and require further investigation to confirm or exclude this possibility.

It is important to note that while previous research has shown the theoretical existence of multiple pathways through which this virus can invade the CNS, some studies suggest that SARS-CoV-2 engages in limited invasion of the CNS. An autopsy study of patients infected with SARS-CoV-2 found typical CNS symptoms, such as hemorrhagic infarction, microglial activation, and neuronal phagocytosis. However, detectable levels of the virus in the brain were very low and might not be associated with histopathological changes [38]. Another study measured SARS-CoV-2 presence in patients’ cerebrospinal fluid using PCR and assessed intrathecal synthesis of virus-associated antibodies. They found that both were rare, suggesting that most neurological complications associated with SARS-CoV-2 are unlikely to be related to the direct viral infiltration of the nervous system [39]. These results indicate that the intrusion of SARS-CoV-2 into the CNS is a rare phenomenon, which should be mentioned in further studies.

## 3. Mechanisms of SARS-CoV-2’s Impact and Damage on CNS

Upon entering the CNS, SARS-CoV-2 can directly affect both neurons and glial cells. Additionally, it can indirectly impact nervous system function through other means.

### 3.1. Direct Damage to Neurons and Glial Cells

Studies have shown that SARS-CoV-2 can infect nerve cells directly. Autopsies of COVID-19 patients have shown that the virus can infect the entire body, including the brain [23]. A clinical study has also found evidence of neuronal and glial degeneration in COVID-19 patients, even in those without significant neurological symptoms. The levels of serum neurofilament light chain (sNfl) and glial fibrillary acidic protein (GFAp) may serve as prognostic indicators for in-hospital mortality related to COVID-19 [40]. This evidence suggests that SARS-CoV-2 may impact neurons and glial cells.

#### 3.1.1. Neurons

To study the effects of SARS-CoV-2 on nerve cells, the researchers used human brain organoids derived from hiPSCs for simulations. Their findings revealed efficient viral replication, which led to neuronal cell death, particularly in cortical neurons. Additionally, infected neurons experienced a loss of excitatory synapses [41]. Another study using human brain organoids confirmed that the virus infected neurogenic cells and manipulated the metabolism and genes of neuronal cells to facilitate viral replication. Furthermore, the authors observed distinct metabolic changes in infected cells compared to neighboring cells, indicating local alterations within the microenvironment that affect nearby cell survival [42]. The antiviral drug Sofosbuvir was found to inhibit the replication of SARS-CoV-2 and reverse these neuronal changes in infected brain organoids [41]. The evidence suggests that the virus directly infects and manipulates neurons, impacting their metabolism, transcription, and surrounding micro-environment, ultimately leading to neuronal damage and death. It is important to note that viral replication within neurons may significantly contribute to these detrimental effects.

In addition, the phenomenon of neuron cells fusion has also been reported in studies of Pseudorabies virus, varicella-zoster virus and herpes simplex virus type 1 [43]. Neuronal fusion may be regarded as a possible pathway by which SARS-CoV-2 interferes with the normal physiological activity of neurons. It has been reported that the S protein of SARS-CoV-2 could induce fusion events between neurons or between neurons and glial cells. This facilitated the transfer of macromolecules or organelles, such as mitochondria, across interconnected nerve cells. Viral-induced neuronal fusion resulted in a gradual amalgamation of surrounding neurons, leading to severe impairment of neuronal activity. Although the affected neurons did not undergo direct cell death, their functionality was significantly compromised [44]. However, compared with neuroinflammation, SARS-CoV-2-induced neuronal fusion is still very rare in general [45]. Further effects on brain function after neuronal fusion caused by SARS-CoV-2 remain unclear. Therefore, clarifying how SARS-CoV-2 destroys neurons requires further investigation.

#### 3.1.2. Glial Cells

In addition to neurons, glial cells are also vulnerable to SARS-CoV-2 infection. The following section will explain the virus’s impact on different types of glial cells. Table 1 summarizes the effects of SARS-CoV-2 on various glial cells.

Although astrocytes do not express AEC2, SARS-CoV-2 can infect them through the NRP1 receptor and replicate in astrocytes within these cells [46,47]. Astrocytes are the support cells for neurons and are also involved in memory formation. Infection of astrocytes with SARS-CoV-2 would disrupt several pathways, including type I, II, and III interferon signaling; retinoic acid-inducible gene 1 (RIG-I); anti-melanoma differentiation-associated gene 5 (MDA5); nucleotide-binding oligomerization domain 2 (NOD2) sensing; and proinflammatory chemokines/cytokines [46]. Infection would also lead to increased oxidative metabolism of astrocytes and decreased neuronal support function. The release of soluble factors reduces neuronal activity and ultimately leads to neuronal death [47]. In terms of memory formation, astrocyte memory formation and storage require normal glucose metabolism and astroglia-neuronal lactate shuttle (ANLS), while the cytokines secreted by astrocytes would act on microglia, and the cytokine imbalance will affect synaptic plasticity and spatial memory [57,58]. Abnormal astrocyte metabolism after SARS-CoV-2 infection may partly explain the brain learning and memory impairments reported during the pandemic [46,48]. Neurodegenerative diseases (such as Alzheimer’s disease, Parkinson’s disease, amyotrophic lateral sclerosis, Huntington’s disease) also have abnormal astrocyte function [59]. Since the relationship between COVID-19 and the above neurodegeneration has not been fully established, further in-depth research and basic experiments are needed, and the changes in astrocytes are a direction worthy of attention. In addition, one important function of astrocytes is their role in maintaining the integrity and function of the BBB. Astrocytes could regulate the permeability of the BBB by influencing the expression of tight junctions in the endothelial cell layer [60]. A clinical study has shown that the serum level of astrocytic calcium-binding protein S100b was associated with increased BBB permeability, while it significantly correlated with COVID-19 disease severity and inflammatory markers (ferritin, C-reactive protein, procalcitonin) [49], which might indicate that the breakdown of the astrocyte barrier may be related to the breakdown of BBB permeability. These experimental facts suggest that there appears to be a relationship between astrocyte and BBB disruption after COVID-19 infection. The increase in BBB permeability may be a consequence rather than a beginning, which deserves further investigation.

The function of oligodendrocytes is to form myelin sheaths [61]. Several reviews and experimental or theoretical studies have suggested that brain demyelination is one of the mechanisms of CNS injury [62]. Animal experiments showed that myelin damage occurred in mice with mild respiratory infection of SARS-CoV-2, leading to a significant loss of oligodendrocytes after infection. Prolonged infection has been observed to deplete the population of oligodendrocyte progenitor cells, potentially compromising neural circuits and axonal health [50]. This study suggests that the effects of SARS-CoV-2 on oligodendrocytes may be manifested in myelin destruction. The formation of new myelin could regulate neural circuits and promote memory consolidation, and correspondingly, the destruction of myelin could affect cognitive function and memory [63]. These findings explain the occurrence of neurological symptoms such as cognitive impairment and memory deficits following SARS-CoV-2 infection. Although the existing evidence partially supports SARS-CoV-2-induced microglial damage, further research is warranted due to insufficient understanding of the relationship between SARS-CoV-2 and microglia. And investigating the mechanisms underlying oligodendrocyte and myelin destruction in COVID-19 patients may provide new insights into other neurological diseases.

Microglia are a type of immune cell in the CNS. The effects of SARS-CoV-2 on microglia are twofold: activation of microglia to induce neuroinflammatory responses and infection of microglia themselves. The S protein of SARS-CoV-2 stimulated the re-release of pro-inflammatory interleukin-1β (IL-1β), interleukin-8 (IL-8), interleukin-6 (IL-6) and matrix metalloprotein-9 (MMP-9) from cultured human microglia through the activation of Toll-like receptor 4 (TLR-4) [51], and also promoted the activation of microglial NOD-like receptor thermal protein domain-associated protein 3 (NLRP3) inflammasomes through the activation of the ACE2-NF-κB axis, leading to neuroinflammation [52]. The virus’ RBD stimulates release of TNF-α, IL-18, and S100B via ACE2 signaling [51]. Microglial nodules have been observed to interact with activated CD8^+^ T cells, contributing to intracerebral proinflammatory effects and coagulation disorders [53]. Furthermore, direct infection of microglia by SARS-CoV-2 occurred via dipeptidyl peptidase 4 (DPP4) binding [54]. RNA-seq analysis revealed that SARS-CoV-2 infection induced endoplasmic reticulum stress and immune response in microglia at an early stage, while late infection caused apoptosis, resulting in a lack of immune response and increased viral replication, leading to neurological symptoms [55]. Microglia also played a role in synaptic ablation. Researchers established a brain organoid model with innately developing microglia and then exposed it to SARS-CoV-2. They observed a threefold increase in microglial engulfment of postsynaptic terminals because the virus activated interferon signaling in microglia and upregulated synaptic elimination genes [56]. This phenomenon suggests that the reduction in synaptic density caused by SARS-CoV-2 infection is most likely achieved by synaptic phagocytosis in microglia. In addition, S protein modulates microglial purinergic signaling, including ATP secretion and upregulation of ectonucleoside triphosphate diphos-phohydrolase (NTPDase)-2 and NTPDase-3 transcripts. In addition, an increase in the transcript and expression of purinergic receptors (P2X7, P2Y_1_, P2Y_6_, P2Y_12_) was observed, which have been implicated in neuroinflammation and neurodegenerative diseases [64]. These receptors and pathways may be associated with post-infection neurological changes and cognitive impairment and deserve to be considered as potential therapeutic targets to treat or alleviate the neurological symptoms of SARS-CoV-2.

In a study comparing the central nervous system effects of the virus during the acute phase of COVID-19 and six months later, they found that plasma concentrations of biomarkers of central nervous system damage (NfL, GFAp) normalized, but a large number of patients continued to have neurological and cognitive symptoms. This study also found that damage to astrocytes occurs early, while damage to neurons occurs later [65]. The findings indicate that damage caused by SARS-CoV-2 to neurons and glial cells in the CNS is limited. Therefore, the emergence of neuro-related clinical symptoms cannot be solely attributed to nerve cell damage. It is important to further investigate other potential CNS effects in future studies.

### 3.2. Damage to the Olfactory System

Anosmia is a typical symptom of COVID-19 infection and is usually the first obvious symptom to appear [66]. The loss of smell after COVID-19 infection can be summarized in two aspects: short-term anosmia in the acute infection phase and long-term post-COVID-19 hyposmia. Mona Khan et al. elucidated the mechanism by which SARS-CoV-2 attacks the respiratory and olfactory mucosa, resulting in anosmia during acute infection. The olfactory mucosa is distributed within the nasal mucosa. SARS-CoV-2 infected the ciliary cells of the respiratory epithelium and the supporting cells of the olfactory epithelium (similar to glial cells), while infection of the supporting cells would result in loss of smell (refer to Figure 1). They also found that the virus could reach the leptomeningeal layers surrounding the olfactory bulb, but the virus did not infect olfactory sensory neurons or neurons in the olfactory bulb [67]. Another study focusing on genetic aspects confirmed that viral targeting occurred primarily in supporting cells rather than olfactory sensory neurons at the receptor level. They found that ACE2 was expressed in olfactory support cells, olfactory stem cells, and perivascular cells, but not in mature neurons [68], indicating that neither olfactory sensory neurons nor neurons within the olfactory bulb were infected. Results from an animal study also support this claim. Researchers used the virus to infect K18-hACE2 mice to study the pathogenesis of anosmia. They found that the loss of smell appeared to be caused by the initial infection and damage to the supporting cells, rather than direct neuronal damage. Plasma therapy alleviated lung infections in mice but did not significantly improve olfactory loss [69]. The results of these studies revealed a mechanism of olfactory loss in the acute phase of infection: the acute olfactory loss was due to damage to the supporting cells in the olfactory mucosa rather than to the olfactory sensory neurons. This was consistent with the fact that olfactory loss first occurred in the early phase of acute infection.

However, another study suggested that this mechanism could not explain the long-term loss or impairment of smell in post-COVID-19 symptoms. They suggested that anosmia could be explained by downregulation of the gas sensing pathway and that nuclear remodeling of olfactory sensory neurons was a potential cause of COVID-19-induced anosmia [70]. First, the authors observed viral infection in hamsters and found a low frequency of virus infecting olfactory sensory neurons, while the duration of anosmia caused by supporting cell infection was too short to explain the prolonged loss of smell in post-COVID-19 symptoms. One possible mechanism of olfactory loss in post-COVID-19 symptoms was downregulation of olfactory receptor gene expression caused by SARS-CoV-2 infection. Olfactory recognition genes were concentrated in the specific genomic compartments (OR compartments) of olfactory sensory neurons [71], while viral infection led to nuclear recombination of olfactory epithelial cells. This process disrupts OR compartments, resulting in olfactory abnormalities. The formation of OR compartments occur during cell differentiation; therefore, their destruction is irreversible in mature olfactory sensory neurons, which may explain why the sense of smell remains absent for a prolonged period after COVID-19 infection. This is consistent with the suggestion in another study that the sense of smell gradually recovers over a long period of time after infection [72].

Overall, anosmia is a dual phenomenon. During acute infection, olfactory loss is caused by damage to supporting cells, while the gradual recovery of olfactory sense after acute infection is due to the involvement of olfactory sensory neurons and nuclear gene remodeling. Acute anosmia is often considered one of the most common symptoms at the beginning of infection, compared to other relatively rare CNS symptoms [73]. However, the incidence of long-term anosmia, which refers to damage to olfactory sensory neurons, is lower [74]. Olfactory dysfunction caused by mucosal damage is more common than direct injury to sensory neurons.

Anosmia caused by SARS-CoV-2 has also been associated with mental health effects. Clinical studies have shown a correlation between COVID-19-induced loss of smell and poor sleep quality, increased fatigue, and increased depression [75]. Animal studies have also shown that mice with olfactory bulb lesions exhibit depression-like behaviors and sleep disturbances [76]. These studies suggest that loss of smell in COVID-19 patients is not only a clinical symptom but also closely related to some psychological manifestations. Anosmia can be considered a typical process by which viruses affect the nervous system, resulting in typical clinical symptoms and adverse effects on patients’ mental health. Further research should focus on whether this virus continues to damage the CNS after inducing a loss of smell and the possibility of developing additional unknown neuro-related symptoms, thereby deepening our understanding of the phenomenon of anosmia.

### 3.3. Vascular and Pericyte Lesions

In a sense, many of the symptoms of COVID-19 can be summarized as constituting endothelial disease [77]. SARS-CoV-2 has the ability to infect endothelial cells and cause endodermatitis [78,79], activate complement [80], induce thrombin production [81], trigger platelet activation [82], and activate the immune response [83], ultimately leading to thrombosis and vascular disease [84]. These classic pathways of endothelial destruction and thrombosis are also frequently observed in other diseases. In addition to the classic endothelial response described above, SARS-CoV-2 can also affect the nervous system through several unique pathways and cause vascular disease.

A study by Jan Wenzel et al. focused on vascular endothelial destruction by the major protease M^pro^ of SARS-CoV-2 and provided new insights into the mechanism of endothelial destruction. The authors first proposed that SARS-CoV-2 infection is associated with an increase in string vessels in the brain. The string vessels are the remnants of dead capillary endothelial cells and manifest as hollow basement membrane tubes that typically contain pericellular protrusions. The researchers then showed that SARS-CoV-2 infection induced brain endothelial cell death via M^pro^ cutting the NF-κB essential modulator (NEMO), which serves as a critical scaffolding protein within the NF-κB signaling center. The inactivation of NEMO prevented its activation of IL-1β and subsequent transcriptional stimulation of NF-κB genes, leading to endothelial cell death, reduced pericellular coverage, increased GFAp expression levels, inflammatory responses, and ultimately far fewer blood vessels within the brain and an inadequate blood supply [85]. Furthermore, they demonstrated that NEMO induces vascular disease via the receptor-interacting protein kinase (RIPK) signaling pathway. These vascular lesions, including string vessels, show a strong association with cerebral ischemia, Alzheimer’s disease, and Parkinson’s disease [86,87,88]. This study provides an alternative perspective on endothelial disease compared to that provided by conventional approaches. The loss of blood vessels and insufficient blood supply caused by SARS-CoV-2 lead to vascular and neurological symptoms, and their long-term effects may also be one of the factors for developing post-COVID-19 symptoms.

Unlike other blood vessels, the capillary endothelium in the CNS surrounds peripheral cells; therefore, the effect of SARS-CoV-2 on these cells is noteworthy. Pericytes play a critical role in vasoconstriction [89]. COVID-19 may also affect brain blood stream perfusion by affecting peripheral cells. S-protein induces the oscillation of Ca^2+^ signals in pericellular cells, changing their morphology from a relaxed circular state to an elongated state [90]. The receptor-binding domain (RBD) of SARS-CoV-2 can act on the ACE2 receptor of pericytes to inactivate it and prevent the conversion of angiotensin II to angiotensin-(1–7) [91]. ACE2 also plays multiple roles in this process. Under conditions of ischemia and hypoxia, the viral increase leads to an upregulation of ACE2 expression in pericerebral cells [90]. These changes induce capillary constriction within the brain. In addition to reducing blood flow, this vasoconstriction also affects blood viscosity at the site of constriction [92], resulting in interrupted blood flow and microthrombosis due to neutrophil blockade. Furthermore, exposure to S-protein induced lipid peroxidation and oxidative and nitrosative stress (anincrease in ROS and RNS) in pericytes. This exposure, accompanied by hypoxia, increased the expression of migration inhibitory factor (MIF), which is an inflammatory cytokine [90]. This unique response of pericellular vascular lesions caused by SARS-CoV-2 in CNS blood vessels deserves further attention.

During the acute infection period, numerous reports have indicated an increased incidence of stroke and other diseases in COVID-19 patients [93,94]. Therefore, it is imperative to conduct more in-depth research and continuous clinical monitoring of the effects of SARS-CoV-2 on cerebrovascular health and blood supply to the brain. In addition, exploring restoration and reconstruction methods for cerebral vessels during rehabilitation and predicting potential disease risks for rehabilitated patients in the future are also worthy of further investigation.

### 3.4. Structure Changes in Human Brain

SARS-CoV-2 not only affects cells and tissues but also disrupts normal brain structures. Imaging studies can demonstrate changes in brain structure after infection with SARS-CoV-2. Figure 2 presents a schematic diagram depicting the potential impact of COVID-19 on brain structures.

SARS-CoV-2 affects the functional connections of the human brain. By observing the spatial distribution of low-frequency power in the brain, researchers can gain insight into its functional organization and connectivity. On this basis, a study was initiated to observe structural abnormalities in the brains of COVID-19 patients. The corresponding researchers measured spontaneous brain activity in hospitalized COVID-19 patients using ALFF (low-frequency fluctuation amplitude of resting fMRI signals). The results showed a decrease in ALFF within the right frontal (middle frontal gyrus) and bilateral temporal lobes (inferior temporal gyrus), while an increase in ALFF within the right parietal (precuneus) and occipital lobes (middle/inferior occipital gyrus) was observed in recovered COVID-19 patients compared to healthy individuals (Figure 2). Notably, a negative correlation was found between the right medial frontal gyrus ALFF and CT score indicating lung involvement [95], suggesting a possible link between frontal lobe abnormalities and hypoxia. Interestingly, more severe right-sided brain damage has been found in studies of other psychiatric disorders (such as insomnia [96], schizophrenia [97], depression [98], and post-traumatic stress [99]). At this stage, these similarities can only be seen as symptomatic similarities; the actual mechanism by which they appear similarly distorted is unclear. Therefore, continued attention and follow-ups are needed to monitor and collect more evidence of structural changes in the brain.

After contracting COVID-19, there are functional abnormalities in the white matter of the brain. A multimodal brain imaging study on long-term COVID-19 patients described the changes in functional brain connectivity and white matter in the patients. Studies have shown a decrease in functional connectivity between the bilateral parahippocampal gyrus, decreased connectivity between the bilateral vermis and orbital superior cortex, and decreased axial or mean white matter dispersion [100]. Previous studies have shown that compensatory hyperconnectivity responses were activated following the disruption of neural connections; however, subsequent depletion of neural resources resulted in a rapid reduction in connections [101]. Additionally, the activation of hyperconnectivity can induce disease and injury [102]. These findings indicate damage to the white matter microstructure of the brain. The abnormal structure of white matter may be caused by acute inflammation, axonal injury, or hypoperfusion, a notion supported by the recovery trend of white matter in patients with COVID-19 [103]. White matter changes represent the brain’s microstructure and have been associated with fatigue severity, reaction time performance, and visual memory [104].

Changes in gray matter were also reported in this study. In terms of gray matter, the volume of gray matter in the anterior cerebellum, parahippocampal gyrus, and frontal, temporal, parietal, and occipital regions was slightly reduced in COVID-19 patients [100]. This reduction in gray matter manifested in a decrease in attention span, memory function, and information-processing speed, which are consistent with previously reported symptoms of cognitive impairment [21]. Another study found an increase in gray matter volume in the brain and significant increases in gray matter in the temporal lobe, insula, hippocampus, amygdala, basal ganglia, and thalamus in both hemispheres [105]. The authors proposed three possible explanations for this symptom: a compensatory recovery effect, a persistent inflammatory response, or a correlation of gray matter volume with duration of infection (with larger gray matter volumes decreasing over time). But they also suggested that the virus had direct effects on gray matter. Overall, the reported changes in brain structure can partially explain the series of events from cellular mechanisms to clinical manifestations. The changes observed in functional connectivity, gray matter, or white matter seem to correlate with the symptoms of cognitive impairment and fatigue response reported in clinical trials. However, further studies and follow-ups are needed to clarify the specific relationship between structural changes in the CNS and symptoms in patients with COVID-19 and to explore possible therapeutic coping strategies.

The hippocampus, a critical component of cognition, is susceptible to damage by COVID-19 (Figure 2). A study of biochemical pathological markers of the hippocampus after SARS-CoV-2 infection showed that hippocampal gray matter atrophy in patients was accompanied by changes in microstructural integrity, hypoperfusion, and functional connectivity [106]. The mechanism of hippocampal injury could be summarized in three aspects: acute injury caused by hypoxia or acute neuroinflammation in the acute phase of viral infection; neuroinflammation and continuous activation of the immune system during convalescence (including damage to and regeneration of neurons); and pathophysiological responses to neurodegenerative diseases occurring in the hippocampus. The pathophysiological responses have been emphasized because alterations in hippocampal structure and function have been associated with cognitive dysfunction [107], particularly attention and memory deficits [108]. Animal studies have also shown that the administration of SARS-CoV-2 S1 protein to the hippocampi of mice induces cognitive deficits and anxiety-like behavior in vivo. These neurological symptoms are associated with neuronal cell death and glial cell activation in the dorsal and ventral hippocampus [109]. Therefore, structural changes in the CNS, particularly in the hippocampus, may be one of the causes of various post-CoV-19 symptoms. In addition, SARS-CoV-2 has been shown to affect neurogenesis [110,111], resulting in morphological changes in pyramidal cells, increased apoptosis rates, a decreased capacity for neurogenesis, and an altered spatial distribution of neurons within pyramidal and granular layers. These effects can lead to memory impairment and may serve as predisposing factors for neurodegenerative diseases such as Alzheimer’s disease. The hippocampus is key to the neurological symptoms caused by SARS-CoV-2. Studies of hippocampal lesions following SARS-CoV-2 infection may help to elucidate the pathogenesis of cognitive impairment caused by SARS-CoV-2 and assess the risk of other neurodegenerative diseases.

Interestingly, mental disorders such as depression, anxiety, and sleep disorders were reported in a subset of long-term COVID patients [112]. Other studies have shown a link between negative emotions and brain structure changes [113,114]. An observational study also noted that the risk of cognitive deficits, dementia, psychosis, epilepsy, or seizures for COVID-19 patients was still increased up to 2 years after being diagnosed [115]. At this stage, the evidence could not prove that the structural changes in the brain were directly linked to mental health after infection, but the relationship between the structural changes in the brain after SARS-CoV-2 infection and the emotional changes and mental health of patients was at least worth investigating. Continued attention and follow-ups are needed to monitor and gather more evidence of structural changes in the brain.

Currently, most research on brain structural changes is conducted in clinical settings. The current results lack systematic generalization, and the current pandemic situation also makes it difficult to establish a COVID-19 patient cohort. More research is needed to elucidate the detailed mechanisms involved. Investigating the effects of SARS-CoV-2 on brain structure and function will provide valuable insight into the specific functions attributed to each region of the brain, thereby bridging the gap between basic research and clinical intervention.

### 3.5. Possible Indirect Affects

In addition to direct effects on the CNS, SARS-CoV-2 may also act indirectly on the CNS after infecting other organ systems. Some possible ways in which COVID-19 infection may indirectly affect the CNS are discussed below.

#### 3.5.1. Lung

The most prominent symptoms of COVID-19 are concentrated in the lungs, continuing symptoms we originally described as pneumonia in the pandemic. However, some studies have shown that lesions in the lungs can affect the CNS. One study showed that exosomes can be transported from SARS-CoV-2-infected lungs to brain regions associated with neurodegenerative diseases. Transcription factors of exosomes regulate neuronal genes, and dysregulated neuronal genes are involved in immune response, signal transduction, and other signal transduction processes that contribute to neurodegeneration [116]. Another autopsy study showed SARS-CoV-2-related brainstem involvement. They found that there were no histopathologic features supporting hypoxemia upon pathologic examination, contraindicating (but not completely ruling out) brain damage caused by hypoxic injury. They also detected SARS-CoV-2 in vagus nerve fibers, which may indicate that the virus can be transported between the brainstem and the lungs, although the direction of transport could not be confirmed in this study. However, this finding suggests that it is possible that pulmonary infection at least affects the CNS [117]. Hypoxia is also a symptom of pulmonary infection, and a small study involving recruited volunteers found that cortical microvascular hypoxia was present in 24% of individuals infected with SARS-CoV-2 but not hospitalized and was associated with clinically relevant chronic fatigue [118]. Clinical studies have also suggested that hypoxemia has a direct effect on neuronal function, which may lead to prolonged recovery of consciousness [119]. Overall, although there is a lack of strong evidence, it can at least be gleaned that pulmonary infection may indirectly affect the CNS and is worthy of further investigation.

#### 3.5.2. Gut and Intestinal Flora

Gastrointestinal abnormalities and intestinal flora disorders are also symptoms that may occur after infection with SARS-CoV-2 [120,121], and some scientists have suggested that intestinal flora could also be used in potential strategies to treat and prevent SARS-CoV-2 infection [122]. Changes in the intestinal flora may also indirectly affect the CNS. At present, there are studies that have analyzed some possible mechanisms affecting the CNS, wherein the main mechanism of action is through the microbiota–gut–brain axis. In the case of an intestinal flora imbalance [123], these mechanisms could be mediated by cytokines or certain chemicals such as gastrointestinal hormones (e.g., CCK), neurotransmitters (e.g., 5-HT), or the secretion of metabolites (e.g., from short-chain fatty acids), causing autonomic nervous system disorders and leaky gut syndrome, ultimately leading to long-term neurological disorders [124]. In addition, another study has shown that ACE2 and TMPRSS2 are expressed in intestinal neurons and glial cells and choroidal plexus epithelial cells of the small intestine and colon, so, histologically, the enteric nervous system and choroidal plexus may be alternative pathways for SARS-CoV-2 nerve invasion [125]. These results indicate that there is some relationship between the imbalance of intestinal flora and CNS symptoms after COVID-19 infection, which deserves further investigation. However, the complexity of the intestinal flora makes it difficult to take these studies any further.

At present, research on the impact of SARS-CoV-2 infection between human systems after is relatively scarce. It is important to recognize that patients with severe neurological symptoms during the COVID-19 pandemic are generally remain a small proportion. Under the current pandemic conditions, it is difficult to find a sufficiently typical cohort of patients for further study. To explore the indirect effects of SARS-CoV-2 infection between body systems, animal experiments are currently a better choice.

## 4. Conclusions and Prospect

The effects of SARS-CoV-2 on the human body are multifaceted and include complications and sequelae beyond those attributed solely to the immune response and inflammatory storm. Among these, the impact on the CNS represents a highly complex aspect. In this review, we summarize the mechanisms by which SARS-CoV-2 invades and destroys the CNS, aside from the immune response and inflammatory storm. Based on the available evidence, we conclude that SARS-CoV-2 may enter the CNS via barrier (BBB and B-CSF-B) crossing or nerve invasion. In terms of CNS damage, the effects of SARS-CoV-2 are manifested at the cellular (neurons and glial cells), tissue (olfactory and vascular systems), and organ (brain structure) levels. While certain hypotheses have been supported by clinical symptoms, it is imperative to acknowledge that numerous ambiguous processes require further investigation and validation.

As the most severe phase of the COVID-19 pandemic has subsided, we have accumulated a substantial amount of data from case analyses and clinical epidemiological studies. The current pandemic phase of the COVID-19 pandemic makes it difficult to establish a large and representative cohort, but this also means that the conditions for conducting pathogenesis research through animal or organoid experiments are more mature and ideal. However, regional outbreaks continue to be reported, new variants are still emerging, and post-COVID-19 symptoms persist in recovered patients. It should be emphasized that although there is much evidence directly or indirectly suggesting that SARS-CoV-2 can invade and destroy the CNS, the vast majority of people will not develop very severe neurological symptoms. Therefore, we should be cautious in conducting further studies and reporting the latest experimental findings and emerging clinical symptoms promptly. At the same time, continuous monitoring and detection of the long-term effects of SARS-CoV-2 on human physiology are essential. The elucidation of the underlying mechanisms of virally induced neurological disorders may allow risk anticipation and facilitate novel therapeutic interventions.

## Figures and Tables

**Figure 1 viruses-16-00663-f001:**
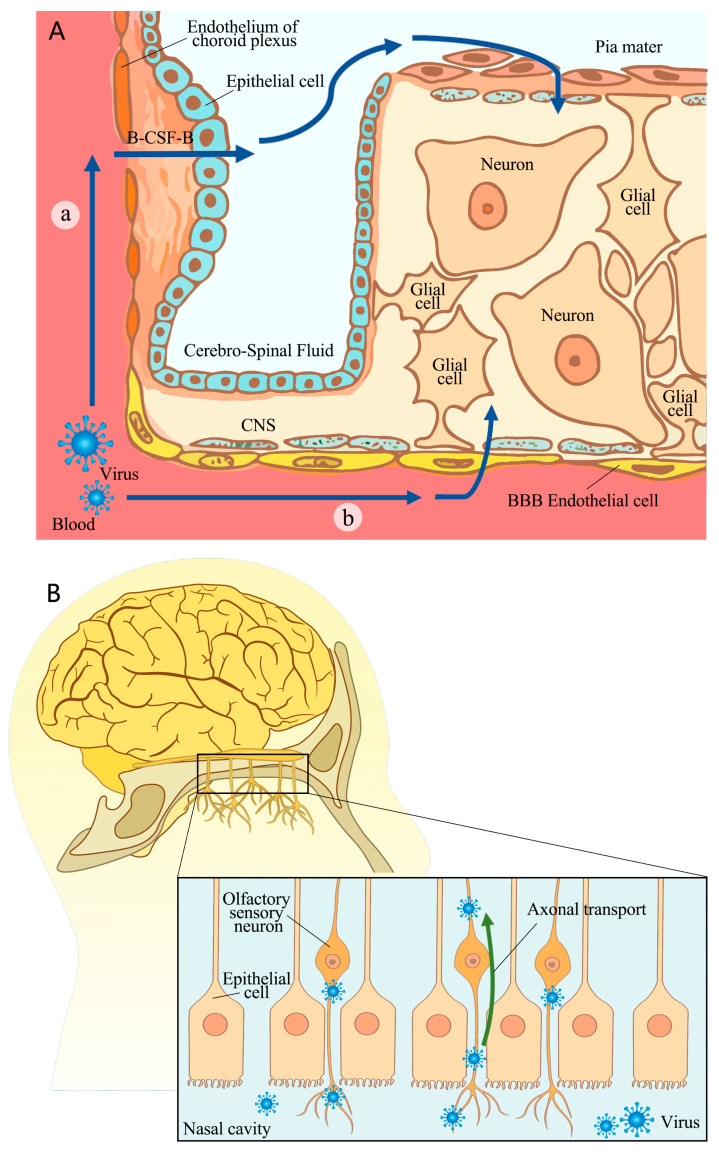
A schematic of possible invasion pathways. (**A**) SARS-CoV-2 may cross the barrier in two ways. (a) B-CSF-B pathway. Blood-borne viruses infect the epithelial cells of the choroid plexus and enter the CNS. The virus enters the epithelium from the basal side, leading to a breakdown of barrier integrity. (b) BBB pathway. Viruses cross the BBB through adsorptive transcytosis mechanisms. The virus enters endothelial cells, followed by intracellular replication, cross-cellular transport, and release viral particles at the basolateral side. Tight junctions were not disrupted during this process. (**B**) The nerve invasion pathway. SARS-CoV-2 invades the CNS by infecting the olfactory nerve. The green arrow represents axon transport. Olfactory cells are located in the mucosa of the superior nasal concha and the upper part of the nasal septum, which allow viruses through the central process of olfactory sensory neurons to the olfactory bulb. This allows SARS-CoV-2 to enter the CNS through neuroinvasive.

**Figure 2 viruses-16-00663-f002:**
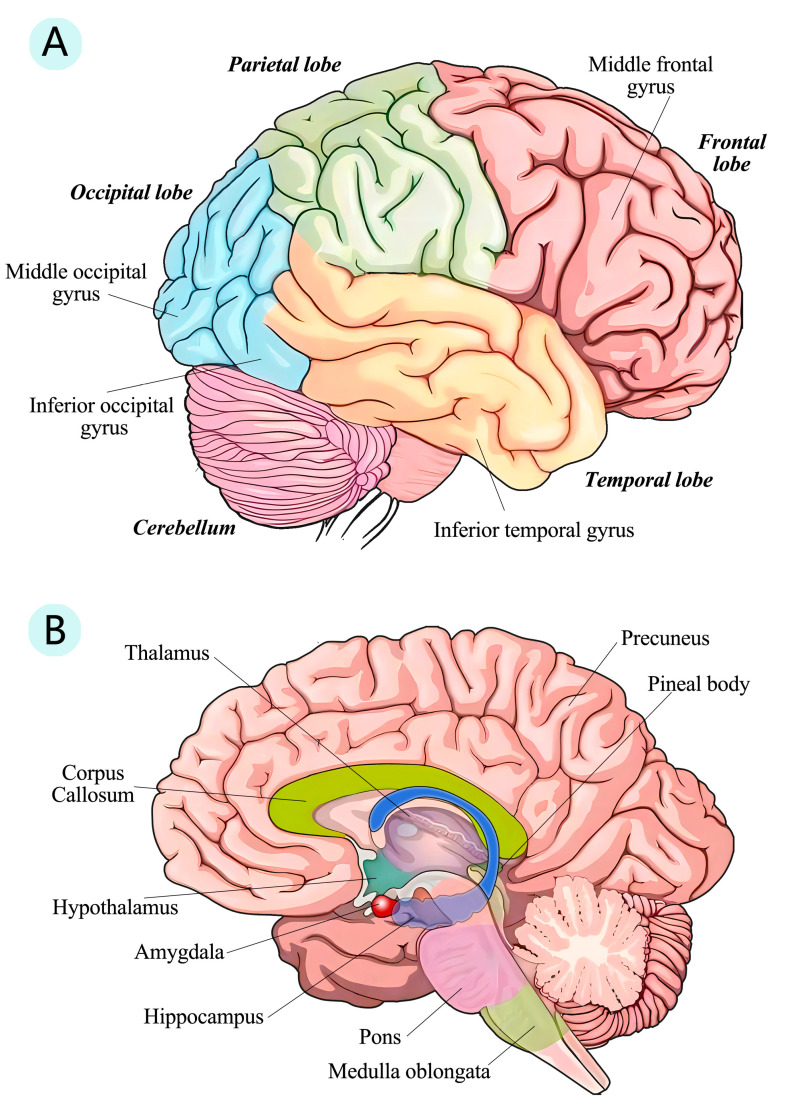
Brain structures reported to be impacted by SARS-CoV-2. The right hemisphere was chosen as an example for illustration purposes. (**A**) Part A displays the lateral surface of the cerebral hemisphere, with annotations indicating the relative positions of brain lobes or gyrus affected by COVID-19. (**B**) Part B displays a midsagittal section of the brain, showcasing the deeper structures within it. This figure highlights the relative positions of the thalamus, hippocampus, and amygdala.

**Table 1 viruses-16-00663-t001:** The effects of SARS-CoV-2 on various glial cells.

Glial Cell Type	Receptor Has Been Identified	Directly Infect Cell	Impact of SARS-CoV-2 Infection	Refs.
Astrocyte	NRP1	√	Disrupts interferon signaling and proinflammatory chemokines/cytokines Oxidative metabolism ↑ Neuronal supporting function ↓ Soluble factors releasing Affects memory formation BBB permeability ↑	[46,47,48,49]
Oligodendrocyte		No direct evidence	Brain demyelination and myelin destruction Compromising neural circuits and axonal health	[50]
Microglia	TLR-4 (Toll-like receptor 4), ACE2	√	Activation of neuroinflammatory responses Synaptic density ↓ (through phagocytosis) Purinergic receptor expression ↑	[51,52,53,54,55,56]

## Data Availability

Not applicable.

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
