# Peer review of "Mechanisms by Which SARS-CoV-2 Invades and Damages the Central Nervous System: Apart from the Immune Response and Inflammatory Storm, What Else Do We Know?"

_viruses, 2024, doi:10.3390/v16050663_

Round 1
Reviewer 1 Report
Comments and Suggestions for Authors
Hi,
My review is written in the word document attached to this message

Author Response
Thank you for your constructive comments for us. The document below is our line-by-line response to the changes.

Reviewer 2 Report
Comments and Suggestions for Authors
The topic is very current and important. The article is written in a good manner, based on a good literature review. Below are my comments to this manuscript.
Lines 45-47
The phrase "rather than solely” is not clear unless there is further reference to some specific theory or research.
Lines 61-64 Figure 1.
The pathways through the BBB and C-CSF-B represent one category, and the nerve invasion route represents the other. However, the way these pathways are presented and described in Figure 1 can be misleading because it would appear that there are in fact three independent pathways: a, b, and c. Moreover, Figure 1 does not show the origin of pathway c - via nerve invasion. The neuron marked with a green arrow could be infected via routes a and b, not specifically via axonal transport.
Lines 65-116
The important differences between BBB and B-CSF-B described in this paragraph provide a good reason to distinguish between these two CNS invasion pathways.
Lines 190-199
The role of astrocytes in the cognitive functions of the brain (learning, memory) has not been clarified. This paragraph can be organized similarly to the section on oligodendrocytes [lines 207-216].
Lines 245-292
This important paragraph requires a slight reorganization. Starting with disorders (anosmia, olfactory impairment), the authors depart from the main topic of the manuscript. The main problem is the knowledge on the mechanisms of invasion and damage of CNS. How research on anosmia and olfactory disorders contribute to this knowledge?
Line 265: “However, this mechanism couldn't explain the long-term loss or impairment of smell.” - Why?
Lines 356-361
Based on one study [63], the authors conclude that structural changes in the brain may be located mainly in the right hemisphere of the brain, and damage to this particular brain area may be associated with many different mental disorders [64-67]. This is an unacceptable simplification both in relation to the various possible changes in the brain after Covid-19 and the various possible neurobiological mechanisms in the mentioned diseases.
Lines 347-420
It is difficult for the reader to understand the detailed results of various studies, conducted using different methodologies, indicating different brain damage after COVID-19. The manuscript would have benefited if the authors organized the whole paragraph according to the specific brain matters and structures that have been systematically described as damaged due to COVID-19 in various studies (using different methodologies). Then associations between these damages and behavioral/cognitive disorders should be presented based on COVID-19 studies when available.
Other mechanisms?
SARS-CoV-2 attacks not only the brain, but many other body organs (including kidneys, liver, lungs, heart, etc.). Dysfunctions of these organs may indirectly affect brain structures and functions. So, could there be additional, important mechanisms of brain damage in people after COVID-19?
Author Response

(The authors gave the same response as above.)

Round 2
Reviewer 1 Report
Comments and Suggestions for Authors
My complete review is in the attached word document

My complete review is in the attached word document
Author Response
Thank you for your comments and suggestions. Please see the attachment below for a point by point reply.

Reviewer 2 Report
Comments and Suggestions for Authors
The authors significantly improved the manuscript in terms of content and language. Work is valuable and necessary. I recommend it for publication.
Author Response
Thank you very much for your kind recognition of our work. We have made some new revisions to the manuscript as requested by another reviewer, and it is now more enriched in content. We believe this more comprehensive manuscript will receive your support. Thank you!